# Iron-Based Magnetic Nanosystems for Diagnostic Imaging and Drug Delivery: Towards Transformative Biomedical Applications

**DOI:** 10.3390/pharmaceutics14102093

**Published:** 2022-09-30

**Authors:** Stefan H. Bossmann, Macy M. Payne, Mausam Kalita, Reece M. D. Bristow, Ayda Afshar, Ayomi S. Perera

**Affiliations:** 1The University of Kansas Medical Centre and The University of Kansas Comprehensive Cancer Centre, 3901 Rainbow Blvd, Kansas City, MO 66160, USA; 2Department of Radiology and Biomedical Imaging, University of California, San Francisco, CA 94143, USA; 3Department of Chemical and Pharmaceutical Sciences, Kingston University London, Kingston upon Thames KT1 2EE, UK; 4Department of Mechanical Engineering, University College London, Torrington Place, London WC1E 7JE, UK

**Keywords:** magnetic hyperthermia, MRI technology, patient-centred healthcare, iron oxide nanoparticles, nanotechnology

## Abstract

The advancement of biomedicine in a socioeconomically sustainable manner while achieving efficient patient-care is imperative to the health and well-being of society. Magnetic systems consisting of iron based nanosized components have gained prominence among researchers in a multitude of biomedical applications. This review focuses on recent trends in the areas of diagnostic imaging and drug delivery that have benefited from iron-incorporated nanosystems, especially in cancer treatment, diagnosis and wound care applications. Discussion on imaging will emphasise on developments in MRI technology and hyperthermia based diagnosis, while advanced material synthesis and targeted, triggered transport will be the focus for drug delivery. Insights onto the challenges in transforming these technologies into day-to-day applications will also be explored with perceptions onto potential for patient-centred healthcare.

## 1. Introduction

### 1.1. Introduction to Iron Based Magnetic Nano Systems

The manipulation of magnetic properties to develop advanced technologies in biomedicine was first used clinically in the early 1980s with magnetic resonance imaging (MRI) as a diagnostic tool [1]. This technology utilized superparamagnetic nanoparticles (SPIONS) as contrasting agents to enhance the MR signals, of which various types of iron oxides were the prime agents [2]. Since then the usage of magnetic iron-based components for imaging and drug delivery has become prominent and steadily expanding areas in biomedical research. Iron is ideal for such applications as it is a ferromagnetic metal (i.e., permanently magnetic) that can form oxides such as magnetite (Fe_3_O_4_) and maghemite (γ-Fe_2_O_3_), which are considered superparamagnetic or display enhanced magnetic properties in the presence of external magnetic fields (Figure 1) [3]. It must be noted that not all forms of iron are magnetic or suitable for biomedical applications. Hematite (α-Fe_2_O_3_) for example, is very weakly magnetic and Wüstite (FeO) is magnetic but is unstable at ambient conditions and is thus nonviable for biomedicinal purposes.

Iron-based magnetic materials have currently become significant in many applications in nanotechnology, including biomedicine [4] and far beyond. The term nanotechnology is defined by the European commission as “…areas of science and engineering where phenomena that take place at dimensions in the nanometre scale are utilised in the design, characterisation, production and application of materials, structures, devices and systems” [5]. This term is rapidly gaining public attention and significant global economic investments due to explosive achievement of technological advances that affect almost every area of life, ranging from electronics [6] such as smart phones and televisions to building materials [7] such as concrete. The materials used in nanotechnology applications, i.e., nanomaterials, can be broadly defined as those that have individual units with at least one Cartesian dimension within 1–100 nm [8]. This definition, however, is not universal and may vary according to the specific field and/or context in which it is used [9,10]. Iron oxide species have been utilized in various formats including nanoparticles, nanofibres [11], sol-gels [12] or as nanoparticle-incorporated composite materials such as hydrogels [13,14,15], core–shell structures [16], liposomes [17,18], etc., which will be discussed throughout this review with respect to materials development and modes and scope of applications. 

### 1.2. Advantages of Iron Based Magnetic Systems

Iron oxide nanomaterials have key advantages that make them particularly attractive for biomedical applications. They have shown to have biocompatibility as proven via many toxicological studies [19] and in vivo applications such as MRI contrast agents for imaging [20], and therapeutic applications such as magnetic hyperthermia and targeted drug delivery, mostly but not limited to cancer treatment [21]. They have been successfully used in multiple clinical trials with hyperthermia based cancer treatment [22,23], that utilizes alternating magnetic fields on a magnetically responsive fluid. This allows for tissue-specific localization of heat in tissues that located deep within the body with high intensity and in a noninvasive manner to treat cancers such as prostate carcinoma and glioblastoma, among others [22], thus illustrating the potential to transform such therapeutics in to safer, more efficient procedures.

Sustainable approaches to synthesis of magnetic iron oxide nanoparticles have seen parallel and rapid growth in recent years, along with biocompatibility studies [24,25,26,27]. One strategy of avoiding or minimizing toxicity in vivo is coating and stabilizing the nanoparticle surface with biocompatible materials such as lauric acid and proteins such as serum albumin [28]. Investigations have also been carried out to customize nanoparticle size, shape, surface morphologies, crystallinity, uniformity, etc. in order to optimize biocompatibility [16,21]. Another promising approach with increasing interest is greener synthesis using biological components and species such as plant extracts, bacteria, fungi and algae. Such techniques are found to be biologically safer and can be achieved in a myriad of shapes such as cubes, tetragonal crystals, spheres, cylinder, and hexagonal rods, etc., among others [29]. Additionally, such biological synthesis has embedded processes that can replace chemicals needed for reduction, capping and stabilization in conventional synthesis, leading to cost-effectiveness [29,30,31,32].

### 1.3. Review Overview

This review aims to discuss key materials and mode of applications of magnetic systems that incorporate one or more nanosized materials containing iron, along two key branches: (1) diagnostic imaging and (2) drug delivery (Figure 2). Insights will be given on the enormous possibilities as well as significant challenges in transforming these materials and technologies for real world applications in biomedicine.

## 2. Diagnostic Imaging

### 2.1. Asset of IONPs to Different Modalities

Iron oxide nanoparticles have ready application towards magnetic resonance imaging as a contrast agent to depress the relaxation rate of tissues where they are accumulated. However, the usage of IONPs can extend beyond its application as a contrast agent for MRI alone. Correctly coupled or functionalized with fluorophores or radiotracers the usage of IONPs can be extended for use in photo-acoustic imaging (PAI), positron emission tomography (PET), and single photon emission computed tomography (SPECT), computing tomography (CT), and magnetic particle imaging (MPI) in addition to its natural contrast agent application in MRI [33]. This can allow for both qualitative and quantitative imaging by combining modalities which enable quantification of iron content in additional to anatomical information.

### 2.2. IONPs as MRI Contrast Agents

Iron oxide nanoparticles are common contrast agents for detecting and tracking cells. They have been shown to retain overall cell viability with rare detrimental effects on proliferation, apoptosis, or necrosis over a wide range of iron concentrations, and can label a wide variety of cell types from stem cells, cancer cells, immune cells, and more [34]. Their adaptability in size, the ability to perform surface modification, and renal clearance capabilities leads to a safer contrast agent that can remain visible for longer periods of time with fewer risks than gadolinium-based agents (GBCA) [35] images. These nanoparticles cause a decrease in the transverse (T_2_) relaxation by creating magnetic field inhomogeneity [34]. This creates signal loss surrounding the iron oxide nanoparticle, commonly referred to as the “blooming artifact,” which is detectable within an image. Comparison of a T_1_ and T_2_ contrast agent can be seen in Figure 3. Iron oxide nanoparticles are viewed indirectly where regions of signal loss vary in size depending upon their target, application, and size of iron oxide nanoparticles used.

### 2.3. Contrast Agent Application in MRI

Iron oxide nanoparticles are incredibly sensitive to the pulse sequence used in image acquisition. These nanoparticles cause a decrease in the transverse (T2) relaxation by creating magnetic field inhomogeneity [36]. This creates signal loss surrounding the iron oxide nanoparticle, commonly referred to as the “blooming artefact,” which is detectable within an image (Figure 4). Regions of signal loss vary in size depending upon their target, application, and the particular size of iron oxide nanoparticle used. Iron oxide nanoparticles are viewed indirectly, where the decreased relaxivity depends on the contrast agent concentration.

In vivo cell tracking can be performed with either an intravenous injection of the nanoparticle or initial incubation with the cells for later injection. For innately phagocytic cells, simple co-incubation will traditionally prompt sufficient labelling; however, for other cell types, efficient labelling may require the use of additional agents such as transfection agents or electroporation [37]. However, upon cell-labelling, it is possible to utilize labelled cells for therapeutic information, such as monitoring the migration and survival of cells and confirming appropriate administration allowing for optimization of treatments to occur in real-time. Labelled cells can also be utilized to image cancer cells with the potential to detect singular cancer cells and monitor metastatic spread. The ability to monitor singular cancer cells is only possible through the blooming artefact making a 20-micron cell present as hypo-intense voids visible in 300-micron slices [36]. A common compound of interest in current clinical trials, Feraheme (Ferumoxytol) is presumed to have a half-life in humans of 24–36 h, allowing a larger window for imaging, and repeat imaging, compared to GBCAs. Ferumoxytol is the leading agent referenced in a majority of clinical trials utilizing iron oxide as an MRI contrast agent. There had been a surge in repurposing this FDA-approved drug initially developed for iron replacement in anemia for chronic kidney disease, yet it had come under scrutiny for off-label use as a contrast agent when a small population of users (not associated with clinical studies for MR imaging) suffered from a series of anaphylactic reactions [38]. Thus, it has now been labeled with an FDA black box warning which initially caused a lull or the termination of ongoing research projects. Fortunately, new safety assessments have indicated that there are less than 1% incidences of moderate to severe toxicities [32] and future research outlooks are promising and excitement over its application has grown.

A study by Corwin et al. investigated the improvement in image quality with steady state MR-angiography (SS-MRA) with ferumoxytol in comparison to conventional first pass MRA. They cited the primary advantage with iron oxide contrast agents was the possibility of administration to patients with renal failure, as with gadolinium-based contrast agents they face the risk of nephrogenic systemic fibrosis. In their analysis they were able to increase the average vessel sharpness significantly, increase resolution, and maintain a comparable signal to noise ratio of the external iliac artery with ferumoxytol assisted SS-MRA with an SNR of 42.2 for first pass MRA and 41.8 for SS-MRA. The use of ferumoxytol provided superior vessel sharpness but also allows for prolonged imaging times, as gadolinium-based contrast agents rapid clearance limits acquisition to first pass -MRA methods [38,39]. While the possibility for high levels of intravascular concentrations of the iron oxide agent can induce artifacts due to the inherent signal lass from a T_2_-based contrast agent, the utilization of lower concentrations can circumvent this shortcoming.

Pan et al. further adapted the iron oxide nanoparticle through the utilization of hyaluronic acid and iron oxide nanoparticles to generate a molecular MRI probe with the ability to adapt and switch from a T_1_ and T_2_ contrast agent for evaluation of atherosclerotic plaques. The overall structure uses HA as the core of the nanomaterials with IONP-P nanoparticles on the surface of the HA platform. These nanoparticles are presumed to by phagocytosed by macrophages where the IONP-HP were found to target macrophages likely due to the CD44 receptor-targeted HA, and form clusters where the particle size of the IONP-HP before clustering is 153 nm and after clustering increases to 562 nm. An initial simulation of the iron oxide nanoparticles and hyaluronic acid nanoparticles (IONP-HP) tested the clusters admission into macrophages in acidic environments with pH’s similar to that of the lysosomes they would inhabit by creating a series of solutions with pH values between 4 to 7.4 and incubated these solutions for a week as they formed clusters. This cluster formation would encourage the T_2_ contrast effect postulated by researchers. TEM results confirmed that when pH levels decrease below 5, the nanoparticles become more compact, and the MRI results confirmed that this produces a T_2_ contrast effect. MR signal demonstrated an increase with incubation time for T_1_-weighted images and a decrease in T_2_-weighted images with the SNR in the T_1_ images decreasing slowly with time with peak signal at 12 h with no T_2_ enhancement happening concurrently. The ability of the nanoparticles to present T_1_ or T_2_ enhancement was found to depend upon the particle size, although the aggregation of the nanoparticles was found to affect both transverse and longitudinal relaxation in MRI [40]. It is postulated that the IONP-HP clusters aggravate the proton dephasing of surrounding water molecules; thus, the T_2_ contrast effect is enhanced by the variety of sizes within the cluster. Animal studies with mice to determine if MRI would be able to identify stable versus vulnerable plaques determined that MRI identification of stable vs. vulnerable plaques was possible upon obtaining T_1_ and T_2_ weighted images. The signal intensity of vulnerable plaques has an apparent increase at 2 h post-injection for T_1_ weighted images, whereas the stable plaque does not, likely due to poor infiltration of macrophages into the stable plaques. This confirms the generation of a “switchable” MRI probe where the IONP-HP can be “turned on” through macrophages as engulfing time increases, thus providing a novel molecular MRI probe for future uses built upon a hyaluronic acid platform.

Current clinical trials include studies such as those performed at the M.D. Anderson Cancer Center which recently completed a clinical trial assessing the use of Feraheme to enable researchers in visualizing cancerous lymph nodes and perform liver imaging with MRI. It involved the intravenous administration of Feraheme at 6 mg of iron/kg and then repeat MR scans over a series of 3 days. The initial scan was taken without the application of the contrast agent and then 38 and 72 h post ferumoxytol injection. Following suite, the Allegheny Singer Research Institute conducted a study to investigate radiotherapy with iron oxide nanoparticles and MRI-Guided Linear Accelerator (MRI-Linac) for primary and metastatic hepatic cancers. Imaging was performed on a 1.5 T instrument to investigate if the use of ferumoxytol for MR-SPION radiotherapy will assist in the detection and avoidance of functional liver tissue to increase the safety of liver stereotactic body radiotherapy. Furthermore, the National Institute of Neurological Disorders and Stroke and National Institutes of Health Clinical Center recently completed a study of in vivo characterization of inflammation with ferumoxytol on 7 T instruments. Researchers focused on assessing if ferumoxytol caused prolonged changes in the brain in MRI, investigated if iron collected in the globus pallidus, and if the iron oxide contrast agent could assist in the identification of inflammation caused by multiple sclerosis. Participants partook in five clinical visits where ferumoxytol was delivered intravenously on the second with the following three including brain imaging on a 7 T MRI and blood draws. These clinical trials are a small segment of the work being done to test and establish iron oxide nanoparticles as a potential contrast agent to enhance and further the diagnostic capabilities of MRI. Future work is likely to build upon these findings as the number of available iron oxide nanoparticles increase, and enhanced surface modification of them allows for more efficient targeting and applicability.

Overall, iron oxide nanoparticles are a promising contrast agent which can allow for increased sensitivity of MRI detection of small lesions and tissue boundaries. There are a broad variety of synthesis routes possible with extensive surface modifications possible that can increase the targetability and sensitivity of these emerging contrast agent alternatives [41]. Through their increased biocompatibility and tunability it may be possible to generate a selective molecular MR probe that would enable enhanced imaging without impacting overall cell viability or exposing patients to potential toxicity allowing for more accurate diagnostic testing and analysis of disease genesis and progression.

### 2.4. Multimodal Imaging Applications

Blood circulation time was investigated by injections of 100 μL of IONP-HP solution into the tail vein of C57BL/6j mice. Blood samples were then collected at various time points as well as prior to injection, and the iron content was monitored through inductively coupled plasma optical emission spectrometry.

Through Equation (1), where ta and tb are the two time points of blood collection and Ca and Cb are the elevated Fe concentration in the circulation at the two time points after the injection of IONP-HP solution, the blood circulation half-life was determined. Overall cytotoxicity and biocompatibility were tested using an MTT assay with RAR264.7 cells and 15 BALB/c mice. Within the mouse group, the 15 mice were separated into two groups where one subset was injected with 80 μL of IONHP-HP and the other 80 μL of saline. Overall behaviors and body weight were carefully observed, and mice were sacrificed with inspection of the heart, liver, spleen, kidneys, and lungs was performed. For biodistribution, a third group of 10 BALB/c mice were injected with 150 μL of IONP-HP in the tail vein, and major organs were harvested at 24- and 48-h time points. The biodistribution of the IONP-HP was analysed through ICP-OES. Overall, there was no death, weight loss, or behavioural changes in mice after injection and H&E staining demonstrated no significant changes to the major organs. There were slight iron increases in the liver and kidneys at the 24-h time point.
(1)$$t_{1/2}=\frac{t_{b}-t_{a}x0.693}{lnC_{a}-lnC_{b}}$$

Initial in vitro MRI studies to examine the relaxation properties of the nanoparticles included obtaining T_1_ and T_2_ weighted images of different variations of the iron oxide nanoparticles, including the hyaluronic acid-iron oxide nanoparticle, with varying concentrations of iron ranging from 0.062 to 2 mm (0.062, 0.125, 0.25, 0.5, 1, and 2 mM) on a 1.5 T scanner. T_1_ and T_2_ weighted images were acquired through a turbo spin-echo sequence with the T_1_ having a variation of repetitions times (TR) from 200 to 600 ms in 100 ms increments and an echo time (TE) of 6.5 ms, while the T_2_ imaging used a constant TR of 200 ms and a variety of echo times from 13 to 91 ms in 13 ms increments. The longitudinal and transverse relaxation times were then determined for each of the nanoparticle variations, as well as the signal-to-noise ratios. The ability of the nanoparticles to present T_1_ or T_2_ enhancement was found to depend upon the particle size, although the aggregation of the nanoparticles was found to affect both transverse and longitudinal relaxation in MRI [15]. It is postulated that the IONP-HP clusters aggravate the proton dephasing of surrounding water molecules; thus, the T_2_ contrast effect is enhanced by the variety of sizes within the cluster.

Mice were separated into vulnerable and stable groups, placed as such depending upon the feeding strategies used in their development. MR imaging of mice in the vulnerable plaque group was used to acquire T_1_ weighted images of the abdomen with a 9.4 T Bruker instrument. Upon intravenous injection of IONP-HP, T_1_ and T_2_ weighted images of both stable and vulnerable plaque groups before and at various time points post-injection. Initially, at the 30-min time point post-injection, the vulnerable plaques showed no signal change. However, between 1- and 2-h post-injection, the vulnerable plaque mice demonstrated substantial T1 enhancement indicating the IONP-HP had been phagocytosed by the macrophages yet maintained a “loose” structure. Upon 24 h and up to 43 h post-injection, the signals became very low indicated that the IONP-HP in the macrophages had entered the vulnerable plaques and caused the contrast agent to move from T_1_ to T_2_ enhancement. Presumably, the switch from T_1_ to T_2_ mode is due to the loose structure of the IONP-HP collapsing in the secondary lysosome of macrophages. Due to the poor solubility of PAA on the surface of IONPs, the NPs form a cluster structure, thereby switching from T_1_ to T_2_ enhancement. Overall, MRI identification of stable vs. while iron oxide nanoparticles have firmly established a space as potential contrast agents for MRI, ongoing work has also investigated their applicability in other regions of imaging. These tactics allow for increased specificity of imaging while retaining the ability to localize regions within anatomical MRI scans. While these tactics often require additional functionalization of iron oxide nanoparticles, they pose a variety of benefits and uses.

Gholipour et al. developed a PET/MRI probe through a biotinylated thiosemicarbazone dextran-coated iron oxide nanoparticle. Biotin was used to functionalize the nanoparticle and increase delivery of the Ga-69 radiolabeled nanoparticles to tumor regions. These nanoparticles were then tested against subcutaneous 4T1 tumors in BALB/c mice. After tail vein administration, biodistribution studies were conducted and found that while there was a sizeable accumulation of the NP in the liver and spleen, the uptake in the tumor was higher than in other organs (excluding liver and spleen). Overall, the tumor uptake of the injected dose was 5.5% dose/g which was determined to be a middle ground when compared to other studies [41]. However, this multimodal nanoparticle comes with the remarkable benefits of being cheaper to synthesis due to the availability of thiosemicarbazone which was used as the chelator and the aldehyde platform which was used in conjugation of both the chelator and biotin targeting agent. When combined with the increased colloidal stability, this work demonstrates the potential application for usages of iron oxide nanoparticles as dual PET/MRI nanoparticles which can be further modified for optimal tumor targeting.

Bell et al. focused on a functionalized nanoparticle to work as a multimodal agent for optoacoustic (OA) and magnetic resonance imaging. Initial studies focused on assessing the use of iron oxide nanoparticles functionalized with indocyanine green (ICG) and Flamma^®^ 774 as agents to induce increased T_2_ relaxation rate for MR imaging while performing as a multispectral optoacoustic tomography (MSOT) contrast agent as well [42]. The FeO-OA dye nanoparticles were then tested in phantoms as both optoacoustic and MR imaging agents where subsequently the most successful was further tested in vivo against subcutaneous U87MG tumors in NCr nude mice [41]. Preliminary evidence from initial testing and phantom studies indicated the ICG capped nanoparticles had negligible contrast agent efficacy in MR imaging while the FeO-774 had representative peaks in MSOT imaging at 780 nm while retaining the relaxation properties of the iron oxide in MR imaging. After intra-tumoral injection of the FeO-774 nanoparticle the nanoparticle provided strong T_2_ contrast in MR scans, remained stable in the mouse model, and remained detectable in MSOT. This study presents an alternative usage to functionalized iron oxide nanoparticles for utilization in multiple image modalities that do not require exposure to radiation.

In an additional study with multifunctional nanoparticles, Peng et al. generated macrophage laden gold nanoflowers which were embedded with ultrasmall iron oxide nanoparticles (Fe_3_O_4_/Au DSNFs). The iron oxide provided T_1_ contrast for MR imaging due to their small size (USPIOs have been shown to act as a T_1_ contrast agent) 38 while the gold NPs could be utilized for CT contrast enhancements. In vivo studies comparing macrophage laden Fe_3_O_4_/Au DSNFs (MA@Fe_3_O_4_/Au DSNFs) and Fe_3_O_4_/Au DSNFs in a breast tumor model of 4T1 implanted subcutaneously into ICR mice indicated improved performance from the targeted Fe_3_O_4_/Au DSNFs. After tail vein injection the SNR of the tumor region as 1.5–1.9 times greater with the MA@Fe_3_O_4_/Au DSNFs compared to the Fe_3_O_4_/Au alone. Furthermore, after CT the MA@Fe_3_O_4_/Au DSNFs reached a summit value of 46 HU, 1.2 times greater than that of the free nanoparticles. Ultimately, Peng and colleagues were able to sufficiently demonstrate the effectivity of a multifunctional nanoparticle utilizing iron oxide and gold to perform as a multimodal probe for both CT and MRI.

While the use of iron oxide nanoparticles for use as MRI contrast agents is a common application there are a variety of imaging modalities where iron oxide nanoparticles can be useful to enhance diagnostic and therapeutic capabilities. The ability to functionalize iron oxide nanoparticles with radiotracers, alternative metal nanoparticles, and targeting agents lends itself to a multitude of applications. Through the use of additional functionalization, iron oxide nanoparticles lend themselves as a convenient mechanism to overcome the shortcomings of each imaging modality independently.

### 2.5. Potential Socio-Economic Sustainability

Medical imaging is inherently expensive. The cost to build, procure, operate, and maintain the instrumentation is often transferred to the patients with the high cost per scan. Additionally, those responsible for the attainment and purchasing of MRI equipment in particular, have denoted their primary concern is patient and operator safety rather than factors which would induce a lower economic footprint or pertain to socio-economic sustainability [43]. This indicates that there is a large area of growth possible in development of advanced instrumentation, since cost is often a secondary concern to its safety and diagnostic capabilities. As the instrumentation and efficacy of contrast agents improve, patient outcomes are likely to improve correspondingly. Little work has been done to thoroughly investigate the cost-effectivity of iron oxide nanoparticles as contrast agents in MRI and other imaging modalities, however a litany of clinical experiments are underway to test FDA approved iron-oxide nanoparticles for alternative uses as a contrast agent. Targeting the cost effectiveness towards utilizing medical imaging for early diagnosis and intervention in patients is an arduous process which involves the quality-adjusted-life years (QALY) gained compared to the increased cost per year along with comparison of a willingness to pay threshold. These factors are subject to fluctuate depending upon the disease in question, patient age, and other comorbidity factors.

While there are studies that investigate the cost and individual impact on imaging in certain disease models, and the cost-effectivity thereof, the global accessibility and sustainability of medical imaging is more nuanced. Pertaining towards MRI, access to this modality of imaging and health management is relegated to the upper middle- and high-income countries, with 90% of the world lacking access [44]. However, there is large opportunity for growth to access these regions allowing for the full implementation of medical imaging through advancing technologies and targeting the applicability of MRI at lower field strengths. Some of the hurdles include developing instrumentation that can be used in an unshielded environment, has less energy requirements, and can be operated and analyzed via telemedicine to truly address accessibility constraints seen presently [40]. Fortunately, work has already begun to address the need for smaller and potentially portable MR spectrometers, with additional investigation into the utilization of ultra-low-field, and very-low-field MRI in these underrepresented demographics. While these instruments are likely to suffer from lower signal to noise ratios and resolutions, they will still present an additional and capable diagnostic tool that will open doors for patients as well as employment for maintenance, management, and operation of the spectrometers. While access to qualified individuals to operate the instrumentation may be difficult initially, networks which will allow for remote training and access, combined with the possibility of harnessing Starlink capabilities to provide internet access in remote regions, may further assist in bridging the gap of accessibility for this diagnostic tool. With the development of more accessible scanners at lower field strengths, it will become more pressing to find safe and effective contrast agents that will better identify areas of malignancies to afford a comparable level of diagnostic efficacy found at higher fields.

## 3. Drug Delivery

Iron-based magnetic components incorporated into various materials and composites are being intensively explored for targeted and controlled release of drugs, towards development of sustainable, robust and efficient platforms. This section will focus on trends in materials development for advanced drug delivery including drug-carrying nanoparticles, composites, fibres and hydrogels with modes of release including magnetic hyperthermia and chemical and thermal triggers (Figure 5). Specific target applications will also be discussed in terms of cancer treatment, wound care and smart devices/technologies.

### 3.1. Advanced Materials

#### 3.1.1. Nanoparticle-Based: Coatings, Ligands and Composite Materials

The surface of SPIONS can be readily functionalised with linkers, receptors, drug molecules, etc., which allows them to be coated with desired drugs for targeted delivery. This method has been investigated extensively for cancer treatment. Tagging the nanoparticles with receptors recognised by cancerous and tumour cells allows for their uptake by these cells, and provides convenient means of using magnetic nanoparticles (MNPs) as vehicles for drug delivery. Targeted drug delivery systems have used a variety of anti-cancer drug carriers, including magnetic iron oxide nanoparticles and natural biodegradable or non-biodegradable polymers [48]. The drugs can either be adsorbed on the surface of these carriers or enclosed inside of them and can be delivered to a given region using an external magnetic stimuli [49,50,51,52].

Doxorubicin is a widely investigated anticancer drug that can be delivered via MNP coating with multiple linker molecules such as citric acid, folic acid, chitosan, etc., in order to optimise MNP stabilisation, dispersion and effective drug delivery, typically characterised via cytotoxicity [45,53]. Doxorubicin has also been used in conjunction with lipid materials and MNPs to form magnetically sensitive lipocomplexes for in vivo drug delivery against CT26 mouse colorectal carcinoma [54,55,56]. Results have indicated superior anticancer activity by the magnetic lipocomplex, along with higher cell uptake, in comparison with standard doxorubicin. 

Chlorambucil, another viable candidate for MNP-based delivery, is used for the treatment of chronic lymphocytic leukaemia, as well as both Hodgkin’s and non-Hodgkin’s lymphoma [57,58]. In a recent study, chlorambucil was incorporated into Fe_3_O_4_ iron oxide MNPs via a chitosan shell and showed increased efficacy in drug release onto cancer cells, compared to the non-complexed drug [59]. Another recent investigation with anticancer drug violamycin loaded onto 8–10 nm iron oxide nanoparticles, have shown increased effectiveness against to MCF-7 breast cancer cell line [53,60], indicating the potential in such SPION-based systems in drug delivery.

SPION-containing composites, i.e., materials with two or more components that display advanced chemical-physical properties to its individual components, are another area of interest for enhanced drug delivery applications with promising results obtained from graphene oxide-Fe_3_O_4_ and γ-Fe_2_O_3_-SBA-15 silica composites [47,53,61] among others. The latter study indicated the formation nanostructures with “rice like grain” consistency that displayed improved ibuprofen release in stimulated bodily fluids. A polymer-magnetic composite consisting of poly(*N*-vinyl-2-pyrrolidone) and Fe_3_O_4_ iron oxide ring-shaped nanostructure carrying doxorubicin have shown good in vivo tumour inhibition under magnetic hyperthermia [41,54]. Moreover, Zn-Al layered double hydroxide (LDH) nanosheets doped with Fe_3_O_4_ MNPs carrying doxorubicin were found to be effective against HepG2 cancer cell line, according to a recent study [49,62]. This material was found to be pH responsive, with more acidic conditions resulting in a larger doxorubicin release profile, thus allowing scope for chemically triggered drug release.

While promising, the composite SPION-based platforms have drawbacks of being expensive, and having complex synthesis, which will affect scale-up and real life applications. Nevertheless, their superior drug release ability warrants further research and exploration onto counteracting such effects.

#### 3.1.2. Hydrogels Systems

A hydrogel is a 3-dimensional polymer network, which has water entrapped within its molecular space [63]. Thus, they are made predominantly of water, but nanoparticles and drugs can be incorporated into the water-polymer matrix for targeted and/or triggered drug delivery. Iron oxide MNPs have been particularly effective in hydrogel-based biomedical systems, ranging from bioseperation and tissue engineering in addition to drug delivery [60,63,64,65]. Recent research have significantly advanced the development of iron oxide MNP incorporated hydrogels, providing viable options to effectively transform drug delivery in to safer, more efficient and sustainable paths [13].

The polymers used to form hydrogels can be natural, synthetic, hybrid or bioinspired and need to be biocompatible for applications in drug delivery. Natural polymers such as gelatin and alginate have been used to develop hydrogels with doxorubicin loaded SPIONs and have indicated cytotoxicity via pH triggered drug release [56]. Dextran-MNP-based hydrogels have also shown to be magnetic and pH stimuli sensitive drug carrier as a dual tuneable drug delivery system [58]. Although highly biocompatible, such natural hydrogel platforms suffer from the difficulty of high scale extraction of polymers from natural sources.

Polyvinyl alcohol (PVA) is a synthetic biocompatible polymer which can be used to produce hydrogels for a wide range of uses including tissue engineering, graphing [66,67] and contact lenses [68]. One of the advantages of PVA as a hydrogel is its use in facile production of cryogels by simply dissolving in water followed by freeze/thawing [69]. A new study investigated the incorporation of acetaminophen and citric acid loaded Fe_3_O_4_ MNPs into various shapes of PVA hydrogels, followed by characterisation by temperature triggered drug release and magnetic hyperthermia experiments [13]. The study also looked at two different shapes, both a disc shape and a hemisphere shape, which is a novel concept (Figure 6). The study showed that there was evidence of a shape selective aspect in the magnetic hyperthermia studies, rendering a novel path for customized drug delivery for wound care applications and has potential for in vivo applications as well. Another synthetic polymer and Fe_2_O_3_ MNP containing hydrogel as also proven to be promising as an anti-inflammatory drug-releasing agent [70] in a recent study. This hybrid hydrogel system consisting of poly (ethylene glycol)-block-poly(propyleneglycol)-block-poly (ethylene glycol) (Pluronic P123), loaded with MNPs was loaded with diclofenac sodium and tested for pH based and temperature induced drug release, and have displayed superior results compared to its non MNP counterpart.

#### 3.1.3. Nano-Fibre Based Materials

Fibres consisting of a polymer base along with MNPs have been investigated for numerous biomedical applications, including drug-carrying vehicles, advanced scaffolds for tissue engineering and bases for wound dressing [71,72,73] (Figure 7). The morphology of such nanoscale fibres resembles biological tissue, and as a result they are of particular interest in biomedicine. They also have high surface area, surface energy and can be organized into porous hierarchical structures, which makes them ideal for cell and tissue adhesion as well as the adsorption of drug molecules. Fibre dimensions can be readily customized to adapt to different applications with techniques such as electrospinning [74,75,76,77,78,79]. Various morphologies including hollow core−shell microparticle encapsulated fibres have been achieved via this method [80,81]. Furthermore, fibre composition have been optimized to incorporate components with biodegradable [82,83] properties and even those that include living tissue [84,85,86]. Scale-up targeting industrial applications of such materials has also been achieved [87].

In spite of the breadth of research available, challenges still remain and prevent translation into real world biomedical applications, ranging from mass production issues to long-term in vivo stability. For instance, lack of control over fibre diameter, pore sizes, and morphological heterogeneities in the fibres produced via electrospinning, have led to reduced cell penetration, which is critical for long-term use as tissue scaffolds [80,82,88,89]. Nanoscale fibre diameters have also shown to be difficult to attain, unlike those in microscale [79] or have resulted in low yields [78]. Moreover, weaker mechanical strength and toxic and/or non-biocompatible components used during synthesis have limited applications in biomedicine [79]. Biospinning, has been used as an alternative technique to produce fibres with greater mechanical strength for scaffolds for tendons or bones [81]. However, this method is hampered by high cost, difficulty in scale-up, longer production times, and lack of customizability. Melt spinning is another alternative and produces fibres by extruding a heated polymer through a spinneret with textural control for cell applications [90,91,92,93]. This too has shown to be non-viable due to high costs on energy and equipment, and limitations in cell-penetration. One solution to address the latter issue was to encapsulate cells via interfacial complexation, which is cheaper [94,95], yet this too was found to be non-scalable and the fibres produced were morphologically heterogeneous, which were significant disadvantages. A recent study detailed a technique based on infusion-gyration to produce polyvinyl alcohol (PVA) and Fe_3_O_4_ MNP incorporated fibres, in a fast and cost-effective manner, with controllable sizes and the potential for scale-up. The fibres produced via this method were also found to be remotely actuated, rendering potential for significant advancements in drug delivery for patient-centred wound care and tissue engineering.

### 3.2. Advanced Applications

#### 3.2.1. Magnetic Hyperthermia for Cancer Treatment

Hyperthermia, a phenomenon broadly defined in biomedicine as elevating tissue temperatures by means of external stimuli beyond normal physiological values, has been used for treatment of various diseases including Rheumatic conditions [96,97] and immunosuppression in management of pain and inflammation [97,98,99]. Cancer treatment however, is by far the most-explored area in hyperthermia-mediated treatments with applications in drug delivery [100,101,102,103]. It compliments conventional treatment methods including surgery, radiation, immunotherapy or chemotherapy. Hyperthermia treatments can be administered across either locally or the whole body, which advantageous for targeted drug delivery [104,105]. Typical temperature ranges used for cancer treatment via hyperthermia fall into two major categories of temperatures higher than 46 °C and temperatures within 41–46 °C.

Nanocomposites consisting of polymer matrices with SPIONS are gaining prominence in hyperthermia-related research on drug delivery [106,107]. They are superior to conventional methods which have drawbacks such as: (1) excessive heating of surrounding tissue causing cell/tissue damage; (2) under-heating of target areas located deep in the body or inside hard bone tissue; and (3) limited heat penetration resulting in recurrent tumour growth or incomplete removal [92,108]. The use of SPIONs ensures convenient thermal triggering of the nanocomposites via external magnetic fields or radiation to convert dissipated magnetic energy into heat, and are hence, an effective source of inducing hyperthermia [92,103]. Additionally, SPION-based delivery systems have advantages that include: (1) easy absorption into cancer cells due to their small sizes; (2) efficient delivery via multiple routes such as injection, liposomes, etc.; (3) ability to be functionalized with drugs or target-specific binding agents to increase selectivity and efficiency of treatment; (4) cost effective and sustainable due to requiring less trigger energy as a result of high heating efficiency.

An effective drug delivery strategy must encompass the above features but go beyond them and be able to interact with complex cellular functions in new ways [93,109]. Moreover, it needs to be biodegradable, biocompatible, and comfortable for patients with minimal adverse effects both during and after drug administration [95,110]. Furthermore, a high drug loading capability is desired, as well as a simple and cost-effective synthesis process [111,112]. Such desirable characteristics can be achieved through the use of SPIONS, which have proven to be effective in delivering a variety of drugs to a specific target in the human body via sustained or controlled release, in recent studies [113,114]. Several clinical studies have been conducted and have shown promise [22,23], however, MNP- or SPION-based magnetic hyperthermia are yet to enter into real world, applications in healthcare. Nonetheless, their beneficial features offer the possibility to develop advanced and multidimensional approaches to non-invasive and precise drug delivery, and hence, have viable potential to effectively transform the field.

#### 3.2.2. Wound Care Applications

Wound care applications that utilize magnetic nanoparticle integrated hyperthermia are emerging as a promising area for enhanced, safer and less invasive drug release to treat surface wounds. These have been used for controlled release of broad spectrum antimicrobials [46,115,116,117], as well as providing various cues for neural regeneration [102,103,118,119]. Additionally, magnetic fields have shown to modulate mechanosensitive ion channels in cells [120] at low frequencies (<100 Hz), providing further evidence for the potential of magnetic systems in wound care. Furthermore, general heating in therapeutic ranges achievable with these approaches have been shown to act as gene expression triggers [118,121], while systems containing MNPs in conjugation with enzymes [107] and synthetic vesicles [108] have shown promise in enhanced wound healing. Hence, it is likely that a multimodal approach that involves heat-mediated drug release together with systems such as magnetic hydrogels would have a number of useful applications in regenerative medicine.

The best wound dressings should serve multiple purposes, including preventing acute or chronic infection, preserving a balanced environment for moisture and gas exchange, absorbing extrudates and blood from wounds, and stimulating cell migration and proliferation, which will aid in wound healing [109,110,112,114,122,123,124,125]. All these characteristics can simultaneously be present in nanofibre wound dressings. Nanofibre dressings enable for both oxygen permeability and wound protection against bacterial invasion owing to their small pore sizes. Furthermore, because of the ease with which chemical and biological molecules can be encapsulated during the spinning process, nanofibres can have wide appeal as viable vehicles for targeted and localized drug delivery [117,126]. In such cases, therapeutic agents have been readily incorporated into nanofibres for controlled and efficient release.

#### 3.2.3. Magnetically Actuated Smart Devices and Microrobots

Actuation of the drug-carrying platform by an external magnetic field (i.e., magnetic actuation), is a scarcely explored area but has enormous potential in biomedicine. Such systems can potentially lead to remote controlled, precise, and safer pathways of drug delivery. A recent study indicates that shape switching magnetic hydrogel bilayers can be used to develop tubular microrobots by coupling a thermoresponsive hydrogel nanocomposite with a poly(ethylene glycol)diacrylate (PEGDA) layer [46]. The magnetic response has been achieved by using graphene oxide or silica-coated superparamagnetic iron oxide nanoparticles dispersed within the thermoresponsive hydrogel matrix, leading to magnetic actuation capability. Results have indicated that such magnetic composite systems can be optimized via shape (ex. helical microrobots) to enhance drug release and motility. Other studies have shown that core–shell ZnNCs encapsulated within mesoporous silica particles to carry and release drugs under magnetic hyperthermia, as remotely-controlled mechanised nanosystem [118,127].

Magnetic actuation can be taken a step further in to the development of smart devices and lead to advances in the rapidly evolving field of micro-robotics with a focus on biomedical applications [45,119,128,129,130,131]. Magnetically actuated miniature robots are able to access obscure regions of the human body and are capable of penetrating and manipulating matter as small as subcellular entities. Research and development of these systems have expanded rapidly over the past two decades due to high potential application in bioengineering and biomedicine [132,133]. While there are various methods of obtaining actuation in miniature robots, magnetic actuation offers a safe but effective approach to remotely control such systems via dynamic magnetic fields. Recent technologies and systems include soft ferromagnetic robot for surgery [133,134], magnetic and optical oxygen sensor for in situ intraocular sensing [135], magnetic microgrippers for biopsy [122,136], magnetic spore-based microrobots for remote detection of toxins [123,137], fluorescent magnetic microrobots for single-cell manipulation [124,138], magnetotactic bacteria swarms for targeted delivery [125,139], and magnetic scaffold to culture cells for tissue engineering [126,140].

A summary of key applications of iron based magnetic nanomaterials and methods of incorporation are depicted below (Table 1).

## 4. Outlook

### 4.1. Challenges

The use of iron-containing nanoparticles can be challenging, although the human body has mastered the balancing act between using iron as co-factor in vital enzymes and massive inflammation caused by reactive oxygen species (ROS). Because iron(II/III) can easily change its redox state, it is typically catalysing or facilitating one electron oxidation- or reduction-chemistry, thus leading to the formation of radicals (Figure 8) [12,74,141]. Complexation with glutathione and a plethora of other (bio)organic (macro)molecules is able to shift the redox transition of Fe(II) to Fe(III) and vice versa over a broad potential range [127,142]. The human body stores iron as Fe(II) in ferritins and hemosiderin to tightly regulate its availability and delivers stored iron throughout the body by transferrin and other transport proteins [129,130,143]. The total amount of iron stored in the human body is 600 to 1000 mg in adult males and 200 to 300 mg in adult females [129,130]. One of the problems with iron-containing nanoparticles for diagnostics and treatment is that the iron concentration that is administered can exceed the total storage capacity of the human body [131,144]. Smaller nanoparticles can be filtered from circulation by means of renal excretion. The threshold for renal clearance is a particle diameter of about 5.5 nm [133], depending on the chemical structure of the nanoparticle. Larger nanoparticles can be excreted by means of hepatobiliary elimination via bile ducts and intestines [145,146]. Nanoparticles that are being taken up by Kupffer cells undergo long-time retention and slow degradation. Depending on the location of the iron-containing nanoparticles and the existence of protective coatings, biocorrosion occurs within 24 to 96 h. If nanoparticle uptake by Kupffer cells is the dominant pathways, iron has to effectively excreted to avoid iron overload. Otherwise, systemic damage is observed. The major iron-induced pathways leading to the release of hydroxyl radicals (HO) and reactive oxygen species (ROS) in the cells, membrane damage, protein damage and aggregation, mitochondrial damage resulting in cytochrome C release and apoptosis, as summarized in Figure 7 [12,95,143]. Therefore, iron overload has to be avoided, resulting in restrictions of using iron-containing nanoparticles as contrast agents in several diseases, such as chronic liver diseases [136]. It should also be mentioned that iron overload (hemochromatosis) from using iron-containing nanoparticles as MRI contrast agents and blood transfusions is additive. Besides genetic mutations, the blood transfusions are considered the major source of iron overload observed in the clinic [147]. MRI imaging is a suitable method to quantitatively detect iron overload that can be treated by means of chelation therapy [148].

### 4.2. Size-Dependence of Iron-Nanoparticle Toxicity

The bioavailability of iron-nanoparticles is strongly dependent on their size [131,144]. Fe-induced apoptosis or ferroptosis are only observed in organs, which can be reached by the nanoformulations. In general, the ROS generated by large Fe-nanoparticles (d > 5.5 nm) is minimal, indicating that cellular uptake (except for Kupffer cells in liver and spleen) is minimal. In sharp contrast, ultra-small Fe-nanoparticles (d < 5.5 nm) cause significant ROS because of their larger particle surface and faster corrosion kinetics, compared to larger nanoparticles. Although many types of ultrasmall nanoparticles induce the generation of ROS (e.g., Au, SiO_2_, and Fe_3_O_4_), only Fe_3_O_4_ (and other Fe-containing nanoparticles) catalyze the Fenton/Haber-Weiss-Reactions [127,131,137], that generate significantly higher concentrations of hydroxyl radicals (HO.) and higher-valent iron-species. These conditions favour lipid peroxidation and, ultimately, ferroptosis. When using ultrasmall iron oxide nanoparticles are utilized for MRI contrast, Fe-toxicity is closely associated with the injection rate. A slower injection rate will provide more time for binding of bioavailable Fe(II) to ferritin. A typical example of Fera-heme that is injected at a rate of up to 30 mg (Fe)/s in the clinic [138]. In mouse experiments, ultrasmall Fe-nanoparticles were preferentially taken up by the heart, followed by liver, spleen, and lung. Uptake lead in all organs to massive ROS-induced inflammation. Whereas effective uptake by liver and spleen could have been anticipated, because the reticuloendothelial system is located in these organs, effective uptake by the heart was somewhat surprising. If the dosing of Fe-nanoparticles was too high, virtually all mice died of heart failure caused by inflammation [139]. Fe-nanoparticle uptake by the lung is mainly caused by the very high available surface and is potentially life-threatening as well. In contract to virtually all other organs, the iron concentration in the mouse kidneys was found to be low. This is a clear indication that iron from ultrasmall nanoparticles is taken up by the surrounding tissue and does, at least initially, not reach the kidneys within 24 h.

### 4.3. Environmental Considerations

The environmental impact of nanoparticles is under discussion. Whereas it is well established that small nanoparticles are a considerable health risk in high concentrations, larger nanoparticles (d > 10 nm) in lower concentrations generally do not pose a significant risk. However, there is not sufficient data on long-term exposure [140]. In our oxidative atmosphere, virtually all nanoparticles undergo oxidation, with the exception of metal oxides and silicates. These oxidation products can be enriched in the environment, for instance in reductive regions, such as fluvial sediments and sludge from sewage treatment plants [152]. It should also be noted that noble metal nanoparticles have slower oxidation kinetics, which allows them to pass through the filters of a sewage treatment plants into the aquifer where they constitute a health risk [142]. Iron and other d-block metal containing small nanoparticles undergo rapid corrosion in the environment, which releases the metals [143,147]. Depending on the pH-conditions in aquifers, metal hydroxide precipitation can occur [144]. It should also be noted that inhaling small nanoparticles can lead to significant lung inflammation, as discussed above [153].

### 4.4. Potential Impact on Real-Life Practices: Probable Trends

There is agreement in the field that we are one to two decades away from the onset of ultra-high field MRI beyond 10 T in the clinic [146]. This field strength will be required for truly contrast-free MRI imaging of most diseases. Until then, the use of contrast agents will be mandatory to enhance signal-to-noise and to shorten the residence time of patients in the MR imager. Compared to gadolinium(III) compounds, which exhibit significant nephrotoxicity [149], Fe-containing nanoparticles can be better managed in the clinic, albeit they also pose some risk of inflammation [147].

In addition to their clinical use as MRI contrast agents, Fe-nanoparticles are magnetic and, therefore, suitable for magnetically aided drug delivery (including chemotherapy), and angiogenic therapy, in which a massive inflammation is caused in the (micro) blood vessels of the tumour microenvironment to cut off the tumour from nutrient and oxygen supply [150]. Ferroptosis is a most promising treatment strategy of virtually all solid tumours that is driven by iron-dependent lipid peroxidation [137,138,151]. Although it appears that the onset of ferroptosis does not require a mitochondrial contribution, lipid peroxidation will heavily influence mitochondrial morphology, bioenergetics, and metabolism [138]. Ferroptosis is favoured in cancer cells vs. normal cells because of the metabolic dysfunction of the former. When combined with targeting strategies of overexpressed clusters of differentiation at the surface of cancer cells, the use of ultra-small Fe-nanoparticles may constitute a promising new strategy to eradicate primary tumours and metastases alike [124,138].

Fe-nanoparticles can also be utilized to target and enhance the effects of hyperthermia. They can be applied in conjunction with A/C-magnetic [151] and radiofrequency (RF)-hyperthermia [154,155] alike. Since a pro-inflammatory tumour microenvironment is required for a successful immune response, which can be triggered by hyperthermia [156], the use of Fe-nanoparticles is potentially synergetic with hyperthermia and are thus likely to be utilized in such applications in the near future.

## 5. Conclusions

Nanotechnology can play a major role in transforming the biomedical field with safer, effective, more advanced, and socioeconomically sustainable materials and technologies. This review focuses on selected recent and significand trends on magnetic nanosystems that incorporate iron-based materials, for diagnostic MRI technology and drug delivery for showcasing this fast-approaching transformation. Key characteristics, advantages and versatility of magnetic iron oxide nanoparticles are emphasised. Principle features of MRI technology, and how it can benefit by the utilization of iron oxide nanoparticles as contrast agents, is critically discussed along with key developments. Novel material development with advances in mode and scope if application in drug delivery are also discussed with respect to iron oxide nanomaterials. Finally, a look at challenges to these applications along with size-dependent iron-nanoparticle toxicity, environmental considerations and potential on real-life practices including probable trends are highlighted.

## Figures and Tables

**Figure 1 pharmaceutics-14-02093-f001:**
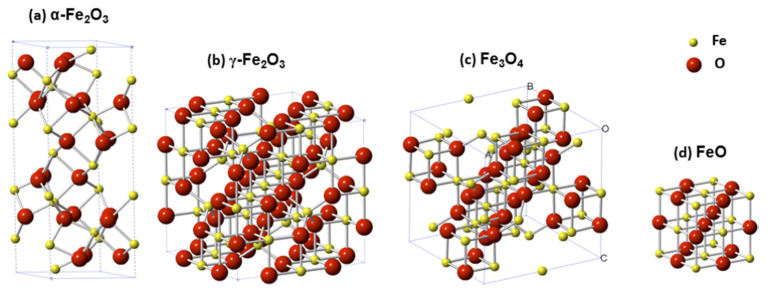
Representation of crystallographic unit cells of four major iron oxides species: (**a**) Hematite—α-Fe_2_O_3_, (**b**) maghemite—γ-Fe_2_O_3_, (**c**) magnetite—Fe_3_O_4_, and (**d**) wüstite—FeO. Image taken with permission from Zhu et al., 2016 [3].

**Figure 2 pharmaceutics-14-02093-f002:**
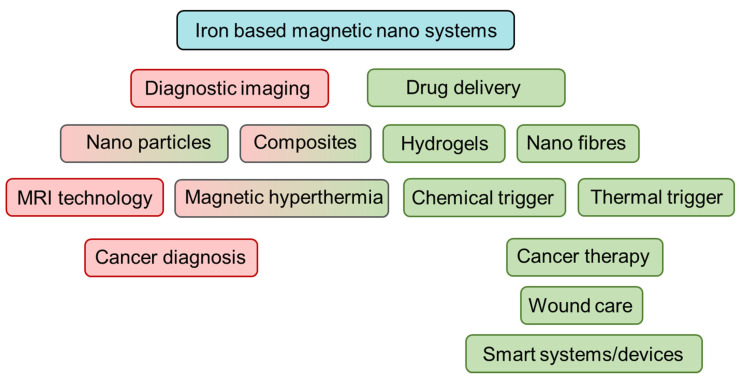
Overview of review topics discussed. Red depicts materials and methods and applications in diagnostic imaging and green represents the same in drug delivery. Boxes with combined red-green indicates both.

**Figure 3 pharmaceutics-14-02093-f003:**
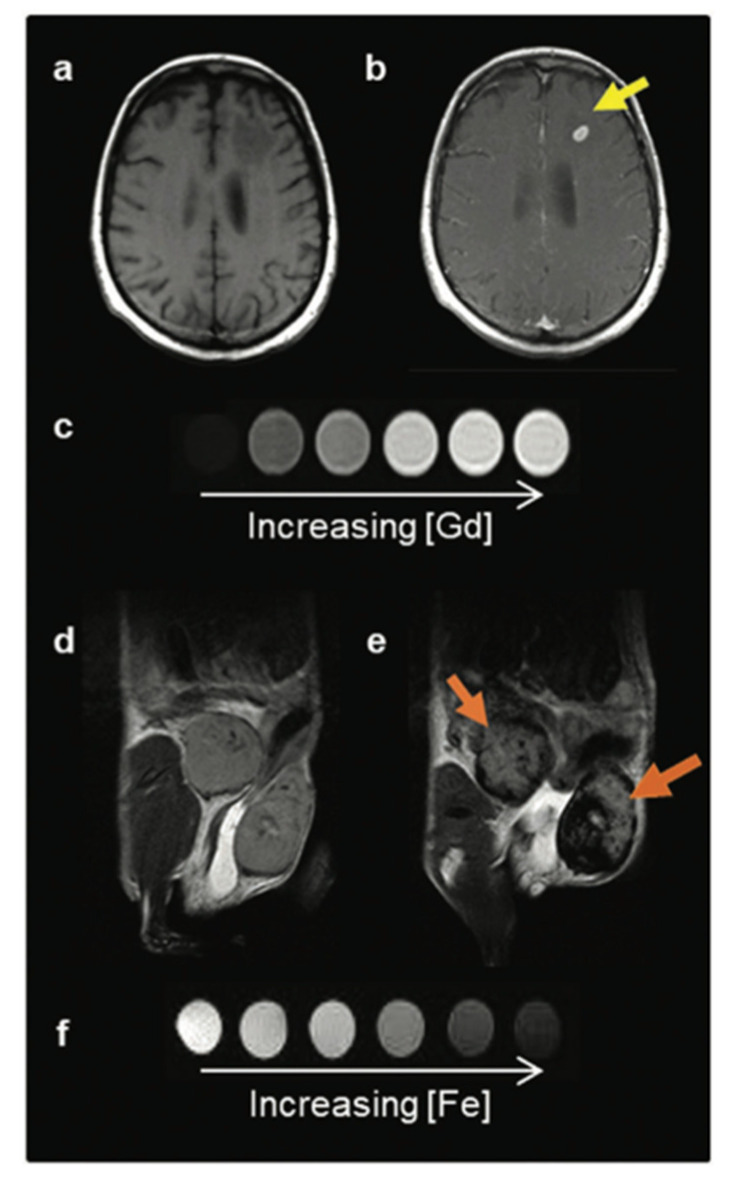
Two classes of MRI contrast agents. (**a**) Pre- and (**b**) post-GBCA T1-weighted MRI on a brain metastasis in a melanoma patient. (**c**) T1 contrast agents decrease the spin-lattice (T1) relaxation time, increasing signal with increasing agent concentration, and produce brighter contrast images. (**d**) Pre- and (**e**) post-IONP-based contrast agent T2-weighted MRI on inflamed mouse mammary gland tumours. (**f**) T2 contrast agents decrease the spin-spin (T2) relaxation time, decreasing signal with increased agent concentration, and produce darker contrast images. Image taken with permission from Jeon et al., 2021 [35].

**Figure 4 pharmaceutics-14-02093-f004:**
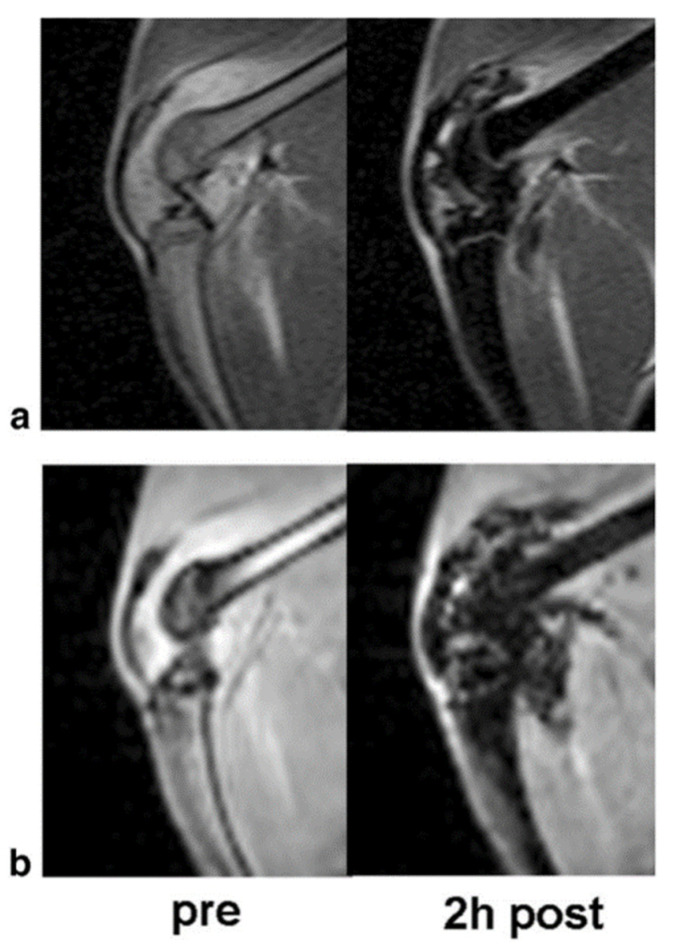
The T2 and T2* effect of ultra-small paramagnetic iron oxide nanoparticles: Corresponding T2-weighted (**a**) two-dimensional Spin Echo (SE) images and T2*-weighted (**b**) three-dimensional Spoiled Gradient Recalled Acquisition in the Steady State (SPGR) images of an arthritic knee joint before and two hours after injection. The USPIO results in hypointense regions with marked signal loss. Image taken with permission from Simon et al., 2006 [36].

**Figure 5 pharmaceutics-14-02093-f005:**
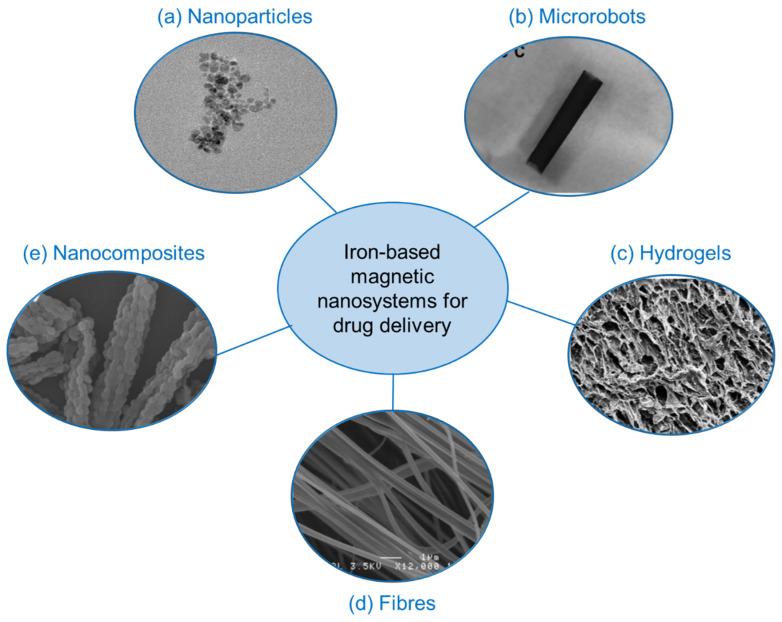
Various advanced platforms consisting of iron-based magnetic nanosystems for applications in drug delivery. Reproduced with permission from (**a**) Guo et al., 2018 [45]. (**b**) Fusco et al., 2019 [46]. (**c**) Perera et al., 2022 [13]. (**d**) Perera et al., 2018 Copyright © 2022 American Chemical Society [11] and (**e**) Huang et al., 2008 [47].

**Figure 6 pharmaceutics-14-02093-f006:**
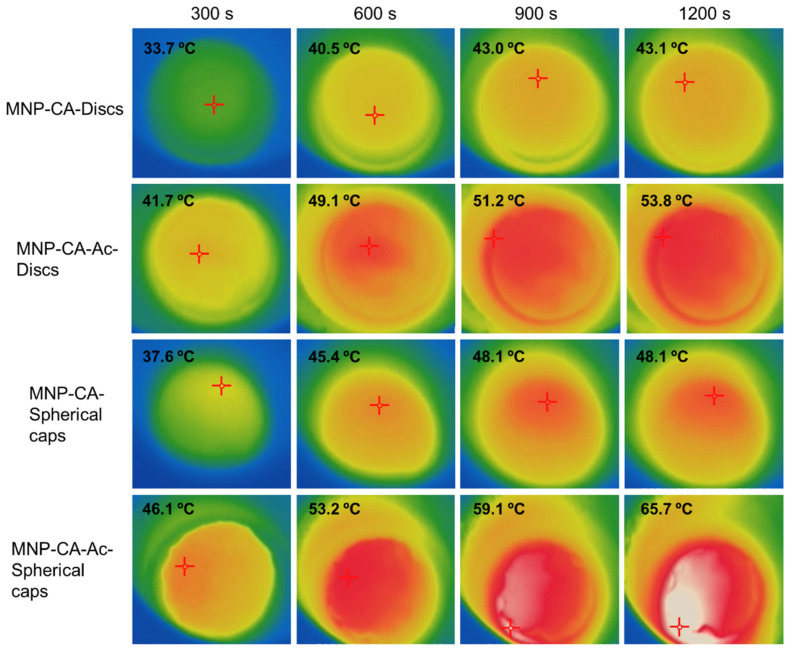
Thermal images indicating magnetic hyperthermia mediated temperature increase of disc shaped and spherical cap shaped gels with and without acetaminophen coating (MNP-CA and MNP-CA-Ac respectively) over time. The hyperthermia set up consisted of a copper coil and a Hikvision ds-2tp21b thermal imaging camera. The coil was heated under 19.95 V with a 14 A (AC) current at a frequency of 0.816 MHz. Gels were kept in the centre of the coil on a Teflon container and subject to heating while thermal images were taken every 10 s for 20 min. Image taken with permission from Perera et al., 2022 [13].

**Figure 7 pharmaceutics-14-02093-f007:**
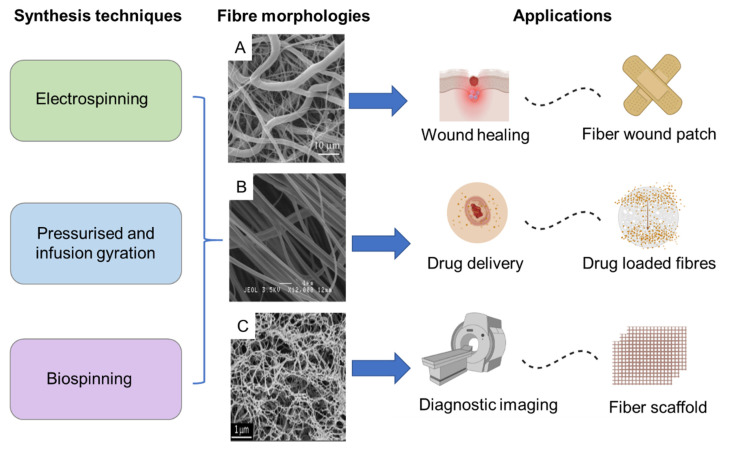
Diversity and versatility of MNP incorporated fibre based magnetic nanosystems in terms of synthesis, fibre morphology and scope of applications. Reproduced with permission from: (**A**)—Do Pham et al., 2021 [72]. (**B**)—Perera et al., 2018 Copyright © 2022 American Chemical Society [11]. (**C**)—Copyright © 2022 Mahmoudi et al., 2016 [73].

**Figure 8 pharmaceutics-14-02093-f008:**
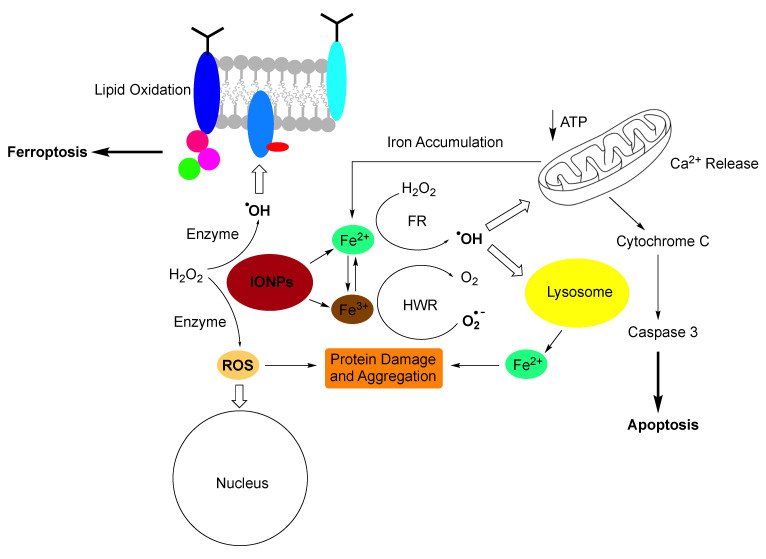
Pathways to iron-induced apoptosis (Ref. [127]) or ferroptosis (Refs. [149,150]). Small Iron-containing nanoparticles (IONPs) create oxidative stress by releasing Fe(II) and Fe(III) cations and by presenting a large assessable surface for redox cycling. In the presence of cellular hydrogen peroxide (H_2_O_2_), Fe(II) enables the Fenton reaction, which created either the hydroxyl radical (HO^.^, shown), or higher valent iron-species Ref. [151] (not shown). Fe(III) is reduced by the superoxide anion (O_2_^−^) in the Haber-Weiss reaction (Refs. [138,151]). Oxidative stress leads to membrane damage, protein damage and aggregation, nuclear damage, mitochondrial damage leading to cytochrome C release and apoptosis (Ref. [150]). Ferroptosis is triggered by radical-induced membrane damage. It is noteworthy that mitochondria do not play a part in ferroptosis (Ref. [137]).

**Table 1 pharmaceutics-14-02093-t001:** Summary of applications, methods of incorporation and examples of iron based magnetic nanomaterials.

Application	Method	Examples	Example Reference(s)
MRI Contrasting agents	Nanoparticles and fluids	Iron oxide nanoparticles and hyaluronic acid nanoparticles (IONP-HP)	Pan et al. [40]
Cancer treatments	Coating	Doxorubicin, Violamycin	Guo et al. [45], Marcu et al. [53]
	Ligand	Chlorambucil-Chitosan Shell	Rozman et al. [57]
	Composite-Coating	LDH-Fe_3_O_4_ (doxorubicin)	Chai et al. [62]
	Composite	Poly(N-vinyl-2-pyrrolidone)-Fe_3_O_4_ iron oxide ring-shaped nanostructure	Wang et al. [48]
	Hydrogels	Dextran-MNP-based hydrogel	Zeng et al. [58]
Wound cleaning	Composite-Coating	γ-Fe_2_O_3_-SBA-15 silica (ibuprofen)	Huang et al. [47]
	Hydrogels	PVA-Fe_3_O_4_ (acetaminophen and citric acid), Poly (ethylene glycol)-block-poly(propyleneglycol)-block-poly (ethylene glycol) (Pluronic P123) hybrid system with Fe_2_O_3_	Perera et al. [13], Pandey et al. [70]
	Fibre	PVA Fe_3_O_4_ MNP incorporated fibres	Perera et al. [11]
Magnetic smart devices and microrobots	Hydrogel-Coating	Hydrogel nanocomposite with a poly(ethylene glycol)diacrylate (PEG-DA) layer	Fusco et al. [46]

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
