# Peer review of "Iron-Based Magnetic Nanosystems for Diagnostic Imaging and Drug Delivery: Towards Transformative Biomedical Applications"

_pharmaceutics, 2022, doi:10.3390/pharmaceutics14102093_

Round 1
Reviewer 1 Report
Please see the following suggestions.
1) Is the reference number is before or after full stop. Please check throughout the manuscript?
2) Section: Drug Delivery: There is another study by Hirak. K patra, 2014 (MRI-Visual Order–Disorder Micellar Nanostructures for Smart Cancer Theranostics). The Fe3O4 nano-carrier for active targeting is designed and used to deliver Doxorubicin via pH sensitive hydrazone linkage and MRI contrast. If possible, you can also add it in your draft as a reference material. It will increase the weightage of the manuscript in terms of doxorubicin delivery via cleavage of hydrazone linkage by magnetic nanocarrier.
3) Line 555: Please correct dis into disc shaped.
4) Line 760: Remove the double gap after reference.
Reviewer 2 Report
The review is interesting, the sections complement each other. However, this review lacks information on the toxicity of nanoparticles (in particular,
the following publications https://doi.org/10.3389/fmats.2019.00179 should be included; https://doi.org/10.1038/s41598-018-25852-4; https://doi.org/10.1016/j.nano.2019.102038 and the like).
A description of the method for searching and analyzing the literature should be included, and for better presentation it would be desirable to include a summary table on magnetic particle applications.
Reviewer 3 Report
The authors propose a review on iron-incorporated nanosystems for diagnostic imaging and drug delivery. This is an interesting topic for a review, and the part on drug delivery is well detailed with a lot of examples.
Nevertheless, I have much more concerns with the part on diagnostic imaging. Indeed, the authors are focusing this part on the use of carbohydrates, which is very restrictive, and not totally linked to the targeted topic of iron oxide nanoparticles. The authors notably mention at page 7, lines 219-224 : « Glycans represent a critical potential component to MRI probes in their ability to increase relaxivity by increasing the overall rigidity of the total contrast agent complex, thus slowing molecular motion. » This is only true for gadolinium complexes and is thus not in relation with the targeted topic of iron oxide nanoparticles. Moreover, the subsequent described contrast agent applications are very limited, since the authors present only 2 articles : The first one is described on nearly 2 pages (pages 7 and 8), which is by far too long with too much details for a review, and the second at page 9 is describing a study involving gadolinium-based contrast agents in a glycosylated silica nanoprobe. This has thus nothing to do with the targeted topic of the review. This whole part has consequently to be written again.
Finally, the part about « potential socio-economic sustainability » at page 11, is too long and not very interesting in the context of the review, especially from line 423 to line 463, since it is not specific of the iron oxide nanosystems but concerns MRI in general.
Reviewer 4 Report
The subject of nanoparticles based on iron magnetic scaffold for theranostic application has gained some interest noweday. The review is well written. I would have some suggestions/comments :
-A paragraph that describe with more details the synthesis of such NP with a scheme would help to understand better the following parts on relaxivity and drug delivery.
-For diagnosis, it would have been interesting to speak about multimodality. Thanks to their specific scaffold, the SPION can hold several complementary modality such as PET/MRI.
-The references are weird sometimes, such as in line 29, 45, 51...
-Some context is missing in the abstract, particularly on which pathologies these iron nanoparticles could be applied.
-The context on the introduction is also little bit difficult to follow. Particularly it would have been interesting to understand the importance of iron NP compared to different NP on the market.
-ln66, for hyperthermia treatment is important to describe the method used (Magnetict, Ultrasound?,...) and the different cancer types associated with (melanoma,...). The authors also disclamed that the NP could be used in other field than cancer. It would be interested to specify little more.
Round 2
Reviewer 3 Report
Globally, the authors have taken into account my remarks but some remaining paragraphs from the first version have to be removed :
- the paragraph from line 367 to line 415 : it refers to the study of Pan et al. This study is described in the previous chapter (contrast agent application in MRI) and has been shortened, which is by far better than in the previous version. Consequently, I don't know why part of the text of the previous version is still present between lines 367 and 415 in the chapter "Multimodal imaging application" since it has nothing to do with multimodal imaging. This has to be suppressed.
- the paragraph from line 480 to line 524 : it refers to the study about glycosylated silica nanoprobes (GSN). As previously mentionned in my previous review, this concerns gadolinium contrast agents and has thus nothing to do in that review about iron oxide nanoparticles. It has thus to be suppressed also.
- This example about GSN has also to be removed from table 1, at the end of the review.
